# Impact of the SARS-CoV-2 Vaccination Program on Presenteeism and Absenteeism among Healthcare Workers in Poland

**DOI:** 10.3390/vaccines12010023

**Published:** 2023-12-24

**Authors:** Karolina Hoffmann, Anna Paczkowska, Michał Michalak, Marzena Jarząb, Wiesław Bryl, Elżbieta Nowakowska, Krzysztof Kus, Piotr Ratajczak, Tomasz Zaprutko, Dorota Kopciuch

**Affiliations:** 1Department of Internal Diseases, Metabolic Disorders and Arterial Hypertension, Poznan University of Medical Sciences, Szamarzewskiego 84 Street, 60-572 Poznan, Poland; 2Department of Pharmacoeconomics and Social Pharmacy, Poznan University of Medical Sciences, Rokietnicka 7 Street, 60-806 Poznan, Poland; aniapaczkowska@ump.edu.pl (A.P.); 8marzenajarzab8@gmail.com (M.J.); kkus@ump.edu.pl (K.K.); p_ratajczak@ump.edu.pl (P.R.); tomekzaprutko@ump.edu.pl (T.Z.); dkoligat@ump.edu.pl (D.K.); 3Department of Computer Science and Statistics, Poznan University of Medical Sciences, Rokietnicka 7 Street, 60-806 Poznan, Poland; michal@ump.edu.pl; 4Department of Pharmacology and Toxicology Institute of Health Sciences, Collegium Medicum, University of Zielona Gora, Licealna 9 Street, 65-417 Zielona Góra, Poland; elapharm@ump.edu.pl

**Keywords:** pharmacoeconomics, COVID-19, vaccines, presenteeism, absenteeism

## Abstract

Background. There is sufficient scientific literature on the effectiveness of registered vaccines in preventing SARS-CoV-2 infection, but research on the impact of the COVID-19 vaccination program on social and economic aspects is lacking. In connection with the above, this study aimed to assess the impact of vaccinations on presenteeism and absenteeism among healthcare professionals in the workplace caused by the COVID-19 pandemic. Methods. A post-marketing, cross-sectional survey-based study was carried out on a sample of 736 actively employed healthcare professionals. Among them, 215 individuals (29.21%) were unvaccinated (control group). The study group consisted of 521 vaccinated respondents, with 52.97% being women and 47.03% men. A self-administered questionnaire was developed and delivered online to the target population group of healthcare workers. Results. A significant association (*p* < 0.01) was observed between the number of doses of the COVID-19 vaccine received and presenteeism among the respondents. Among the unvaccinated respondents (2.30 ± 1.19) or those vaccinated with only one dose of the SARS-CoV-2 vaccine (2.16 ± 1.11), the COVID-19 pandemic had a significantly higher impact on work performance compared to individuals vaccinated with three doses of the vaccine (1.19 ± 1.11). Moreover, a significant association was found (*p* = 0.0265) between the number of workdays missed (over the last twelve months) due to COVID-19-related sick leave and the number of doses of the COVID-19 vaccine received. The number of workdays missed due to COVID-19 sick leave was lowest in the group vaccinated with three doses (2.00 ± 6.75) and highest in the unvaccinated group (5.32 ± 16.24). Conclusions. Our results clearly show that the widely implemented national COVID-19 vaccination program brings tangible benefits both in medical and economic terms. The extent of reducing absenteeism and presenteeism caused by the coronavirus disease depended on the number of vaccine doses administered.

## 1. Introduction

The outbreak of the pandemic caused by the SARS-CoV-2 virus, which spread worldwide, has resulted in serious health, social, and economic consequences. According to data from the statistical portal Worldometer, over 6.8 million people have died due to COVID-19 worldwide. [1]. The fight against the COVID-19 pandemic has necessitated changes in healthcare organization. The Polish government allocated a total of EUR 11.2 billion to the pandemic response [2]. Identifying and understanding the virus’s mechanism has enabled the development of effective therapeutic strategies, but the time required to equalize forces in combating this infectious disease has resulted in costs that have left their mark beyond the realm of health. The pandemic significantly impacted the global economic situation, leading to disruptions in production and services, along with worker quarantines, causing an economic downturn. Poland’s Gross Domestic Product (GDP), which serves as a measure of a country’s economic activity and well-being, decreased by 2.8% in 2020 compared to the previous year [3]. As a result of the COVID-19 pandemic, irreversible changes and costs have particularly affected the healthcare sector. Economic classification of healthcare costs often distinguishes direct, indirect, and intangible costs [4].

Direct costs include expenses related to the consumption of resources necessary for providing medical care. The opportunity to normalize the health, social, and economic situation lies in the development and approval of vaccines within the framework of preventive and protective strategies against the virus. The effects of the disease are not limited to its direct impact on the patient, as they also indirectly affect participants in the economy. Society bears costs due to economic losses caused by the disease [5]. Production loss resulting from the absence of a sick employee (absenteeism), decreased work efficiency of an ill person (presenteeism), informal care costs, and unpaid work are components that contribute to the indirect cost of the disease.

Permanent disability caused by illness and premature death result in economic consequences due to the reduction in Gross Domestic Product (GDP) [4]. In 2020, during the outbreak of the pandemic caused by the SARS-CoV-2 virus, there was a significant increase of 44.7% in medical leave certificates issued in March compared to February of that year, which correlates with the beginning of the pandemic in Poland. From the beginning of the pandemic in Poland, in March 2020 until the end of that year, 642.3 thousand medical certificates were registered for a total of 5068.6 thousand sick days related to COVID-19. Estimated expenses for sick leave benefits in 2020 associated with COVID-19 amounted to approximately EUR 0.5 billion [6].

The pandemic has resulted in economic consequences, manifesting in a 2.7% decline in Poland’s Gross Domestic Product (GDP) in 2020 compared to 2019 (an average reduction of 5.5% among OECD countries). It was the first GDP decrease since 1991. In 2021, the Polish economy began to recover to pre-pandemic levels, but another wave of infections in the second half of the year slowed down the economic progress, leading to a GDP growth of approximately 5% compared to the previous year [7,8].

Vaccination is an artificial, active immunity involving the introduction of a vaccine into the body—a biological preparation containing antigens of disease-causing microorganisms that stimulate the production of specific humoral or cellular immunity [9]. In Poland, the national SARS-CoV-2 vaccination program includes the use of five vaccines: BNT162b2 (Pfizer-BioNTech, Mainz, Germany), mRNA-1273 (Moderna, Cambridge, MA, USA), ChAdOx1-S (Oxford-AstraZeneca, Gaithersburg, MD, USA), Ad26.COV2.S (Johnson Pharm, New York, NY, USA), and Nuvaxovid (NVX-CoV2373) Novavax, Gaithersburg, MD, USA. These vaccines differ in their mechanism of action against COVID-19 [10,11]. With the decision of the Minister of Health from 2 November 2021, the administration of booster doses began for all individuals over 18 years old who have completed their full vaccination schedule. According to recommendations, after receiving the primary vaccination with Comirnaty, Spikevax, or Vaxzevria, the administration of a booster dose with an mRNA vaccine is possible at least five months after the primary vaccination. For those who received the single-dose Janssen COVID-19 vaccine, a booster dose of an mRNA vaccine or Janssen COVID-19 vaccine is recommended at least two months after the primary vaccination [12].

As of 1 October 2022, the National Vaccination Program introduced updated vaccines for the Omicron variant, which were adapted to better match the circulating variants of SARS-CoV-2 [12]. These vaccines may provide increased protection against new variants, especially the Omicron variant and related variants. They include the mRNA of the original SARS-CoV-2 strain and the mRNA of the Omicron variant (subvariants BA.1 or BA4/BA.5). The available vaccines include Spikevax Bivalent Original/Omicron BA.1. (Moderna), Comirnaty Original/Omicron BA.1. (Pfizer-BioNtech), and Comirnaty Original/Omicron BA.4/BA.5. (Pfizer-BioNtech) [13,14]. There is sufficient scientific literature on the effectiveness of registered vaccines in preventing SARS-CoV-2 infection, but research on the impact of the COVID-19 vaccination program on social and economic aspects is lacking. In connection with the above, this study aimed to assess the impact of vaccinations on presenteeism and absenteeism in the workplace caused by the COVID-19 pandemic. The study conducted a comparative analysis of the effectiveness of available vaccines in the Polish healthcare system in reducing employee absenteeism in the workplace caused by the COVID-19 pandemic.

## 2. Material and Methods

### 2.1. Study Population

#### Sample Composition

This post-marketing trial had been designed as a cross-sectional survey-based study. The study group consisted of doctors, pharmacists, nurses, and physiotherapists. Three thousand e-mails were sent to the above healthcare workers with a request to join the study voluntarily. The e-mail contained a link to the research questionnaire and all necessary information about the study’s purpose and rules of participation. Moreover, potential respondents were able to download the link to the study questionnaire from the Poznan University of Medical Sciences website promoting our study project and social media (Facebook, LinkedIn). A total of 1095 respondents (all of those agreed to participate in the study) were taken into consideration. However, based on the inclusion criteria of the study and incomplete questionnaires by 359 healthcare workers, 1130 respondents were finally included in the study. The response rate, defined as the number of adequately completed online forms, was 36.5%. Moreover, 736 healthcare professionals participated in the survey. Among them, 215 individuals (29.21%) were unvaccinated. The study group consisted of 521 vaccinated respondents, with 52.97% being women and 47.03% being men. Participants were selected based on inclusion criteria, including age of consent, place of residence in Poland, giving voluntary and informed consent to participate in the study, access to the Internet, being vaccinated with at least one dose of currently available SARS-CoV-2 vaccines, and being actively employed. The exclusion criteria were as follows: age under 18 y., place of residence outside Poland, lack of consent to participate in the study, no Internet access, not being vaccinated against COVID-19, and not being actively employed. Participants with reduced immunity and individuals with prior clinical or microbiological diagnosis of COVID-19 were excluded from the study.

The survey process began following an opinion (No. 25/22) issued by the Bioethics Committee at the Poznan University of Medical Sciences confirming that the study does not have the features of a medical experiment. Before taking the survey, each participant was made aware of the purpose of the research and was informed that the study was safe, free, and anonymous, and that the consent to participate in the research could be withdrawn at any time, without giving any reason. All obtained information was treated as fully confidential, and the identity of all participants remained anonymous. No personal data of respondents were collected, ensuring compliance with the Personal Data Protection Act, as no data allowing subsequent identification of participants were used in the research project.

### 2.2. Research Methods and Time Horizon

The questionnaire was developed following a standardized questionnaire—The Work Productivity and Activity Impairment (WPAI) [15,16], which measures absenteeism and presenteeism—focused group discussions, and an expert opinion. The study questionnaire was accepted by a group of 5 national medical consultants specializing in the field of infectious diseases.

The questionnaire comprised 20 questions divided into four categories:Demographic data (age, sex, height, weight, profession, and geographic region);COVID-19-related anamneses (type of vaccine, number of vaccine doses, dates of vaccine doses, previous infection, and diagnosis date);Economic evaluation of the SARS-CoV-2 vaccination (impact of the COVID-19 pandemic on work performance and number of workdays missed due to sick leave caused by COVID-19 infection over the past twelve months).

The primary outcome was the impact of vaccinations on presenteeism and absenteeism in the workplace caused by the COVID-19 pandemic. The above assessments were made based on respondents’ answers to the following questions: (1) Over the past twelve months, how many days of work did you miss due to the COVID-19 pandemic? Please provide the number of days taken off due to sick leave or confirmed coronavirus infection. (2) Over the past twelve months, to what extent did the COVID-19 pandemic affect your productivity at work? Please recall days when your work was limited in scope or type, when you achieved less than you wanted, or when you couldn’t work as diligently as usual. Respondents assessed this questions on a scale from 1 to 5, where 1 means “The COVID-19 pandemic had no impact on my work”, and 5 means “The COVID-19 pandemic completely prevented me from working”.

The study was conducted from November 2021 to February 2022. Due to government restrictions resulting from the pandemic and to minimize the risk of SARS-Cov-2 infection among respondents, the survey was conducted using the Computer-Assisted Web Interview (CAWI) method [17,18]. All study questionnaires were distributed through social media. A research questionnaire with a request to fill it in electronically was sent to healthcare workers inviting them to take part in the study. To prevent multiple submissions from the same person, the Google Forms platform recognized the IP address of the interviewer’s computer. The study questionnaire primarily consisted of mandatory fields, ensuring that no answers were omitted. 

### 2.3. Statistical Analysis 

According to the Central Statistical Office (GUS), we have N = 333,876 registered employees with the right to practice one a medical profession. We assessed the sample size using the finite population correction and assuming the estimated proportion in the population of *p* = 0.5, a z-score of 1.96 for the 95% confidence level, and a margin of error (MOE) of 3%. Taking into account the MOE = 3%, the required sample size was n = 1063 responders. However, due to poor response rate of n = 736, we have a sample size which satisfies MOE at the 4% level (required sample size pf 559). 

Counts and percentages were utilized to present categorical variables. As the numerical data did not follow a normal distribution (determined by the Shapiro–Wilk test), they were presented as median and interquartile range Me, [Q1–Q3]. The comparison between the vaccinated and unvaccinated responders was conducted using the non-parametric Mann–Whitney test. Additionally, a comparison among four types of vaccines (BNT162b2 vs. mRNA-1273 vs. ChAdOx1 nCoV-19 vs. Ad26.COV2-S) as well as between the respondents with a specific number of vaccinations (0 being not vaccinated and 1, 2, or 3 vaccinations) was performed using the Kruskal–Wallis test with Dunn’s post hoc tests. The results of the post hoc analysis are denoted in tables by indexing the medians with the letters a or b. Groups denoted by the same letter do not differ statistically significantly. 

To assess the comparison between two categorical features, the chi-square test of independence was used. A *p*-value of less than 0.05 was considered statistically significant for all tests. The statistical analysis was carried out using Statistica data analysis software v. 13.3 (TIBCO Software Inc. 2017, Palo Alto, CA, USA; http://statistica.io (accessed on 15 March 2022)).

## 3. Results

### 3.1. Study Group Characteristics 

There were 736 surveyed healthcare professionals. The comparative characteristics of all the respondents are presented in Table 1. The unvaccinated subjects statistically lived more often in villages and smaller cities (less than 100,000 inhabitants, *p* < 0.0001). For the final study group, 521 vaccinated subjects were recruited, including 276 women and 245 men. The analyzed study groups, i.e., the unvaccinated and vaccinated respondents, did not significantly differ in terms of key socio-clinical characteristics, such as gender, age of participants, and level of education.

### 3.2. The Impact of the COVID-19 Vaccines on Presenteeism

Active healthcare professionals were asked to assess the impact of the COVID-19 pandemic on their work performance. The ratings were based on a scale from 1 to 5, where 1 indicated no impact of the pandemic on their work performance and was chosen by 293 individuals (39.81%). On the other hand, 5 represented a complete inability to work due to the outbreak and duration of the pandemic, selected by 4.07% of the respondents.

In detail, the COVID-19 pandemic had no impact on work performance for 163 individuals (22.15%), while 170 respondents (23.10%) were unsure and chose option 3. Additionally, 80 respondents (10.87%) stated that the pandemic had an impact on their work productivity.

Based on the conducted observations, a significant association (*p* < 0.01) was observed between the number of doses of the COVID-19 vaccine received and presenteeism among the respondents (Table 2). Among the unvaccinated respondents, the median and [Q_1_–Q_3_] was 3 and [2,3,4,5], and among those vaccinated with only one dose of the SARS-CoV-2 vaccine, it was 3 and [1,2,3,4,5]. The COVID-19 pandemic had a significantly higher impact on work performance compared to individuals vaccinated with three doses of the vaccine: 1.5, [1,2,3] (Table 2).

As a result of the conducted research, no association was observed between the type of COVID-19 vaccine received and the impact of the pandemic on the level of work performance among the surveyed respondents (*p* = 0.1313) (Table 3).

### 3.3. The Impact of the COVID-19 Vaccines on Absenteeism in the Workplace

As part of the study, the survey participants declared the number of workdays missed (over the past twelve months) due to sick leave caused by COVID-19 infection. A significant association was found (*p* = 0.0265) between the number of workdays missed due to COVID-19-related sick leave and the number of doses of the COVID-19 vaccine received. The number of workdays missed due to COVID-19 sick leave was lowest in the group vaccinated with three doses, with the median and [Q1–Q3] being 2 and [1,2,3,4,5,6,7,8,9,10], and the highest in the unvaccinated group, with 6 and [5,6,7,8,9,10,11,12,13,14,15,16,17,18,19,20,21,22,23] (Table 4).

On the other hand, it was observed that the type of COVID-19 vaccine received did not have a significant impact (*p* = 0.6531) on the number of workdays missed (over the past twelve months) due to sick leave related to COVID-19 infection among the respondents. The number of workdays missed due to COVID-19 sick leave was lowest in the group vaccinated with the Moderna vaccine, the median and [Q_1_–Q_3_] being 2 and [1,2,3,4,5,6,7,8], and highest in the group vaccinated with the Pfizer/BioNtech vaccine, with 5 and [2,3,4,5,6,7,8,9,10,11,12,13,14,15,16] (Table 5).

## 4. Discussion

The economic value of the COVID-19 vaccine to society consists of the value resulting from minimizing new cases of the disease and the value of rapidly resuming productive activities that form the foundation of the economy.

The effectiveness of implementing the COVID-19 vaccination program can also be expressed from a pharmacoeconomic perspective, considering its impact on indirect costs caused by the pandemic, such as employee productivity during work or the number of days of sick leave due to COVID-19 infection. These markers allow for a partial assessment of the cost-effectiveness of vaccination from the state’s financial perspective. The global COVID-19 pandemic disrupted daily functioning, significantly affecting sectors unrelated to healthcare. The scale of infections and the consequences of the coronavirus disease had a negative impact on the labor market, destabilizing the global economy by reducing worker productivity or causing complete absence from work. Therefore, when assessing the total cost of the pandemic, one must not forget the price paid by society not only in terms of life and health but also the economic expenditure arising from medical and non-medical costs, which continue to have significant financial consequences to this day. The introduction of vaccinations against SARS-CoV-2 allowed achieving herd immunity and presented an opportunity to limit the costs of the pandemic [19,20].

In the presented research, both respondents who were vaccinated and unvaccinated and declared professional activity were asked to answer a question regarding the impact of the COVID-19 pandemic on their work productivity. Assessing individuals’ work productivity is ambiguous, and defining standards describing it is difficult. In the context of the pandemic, it can be associated with the phenomenon of presenteeism. It should also be noted that in many places, employees switched from traditional on-site work to the previously less popular remote mode of work, i.e., “home office.” In the study by Kittgawa [21] analyzing four companies, the impact of remote work enforced by the pandemic resulted in a decrease in employee productivity. Similar results were obtained in Morikawa’s study [22]. On the other hand, Emanuel and Harrington [23] observed an increase in work productivity during the home office mode. These studies referred to the situation at the beginning of the pandemic when vaccines were not yet available.

In the Polish study, work productivity was assessed for all respondents, regardless of their vaccinated status. The majority of respondents stated that the pandemic had no impact on their productivity (39.81%), while only 4.08% of respondents declared a complete inability to perform professional work due to the pandemic situation.

The production and distribution of vaccines contributed to an increase in the productivity of the working population. Administration of the vaccine may temporarily lead to loss of productive time due to its associated adverse effects, which can be compensated in the long run by the productive time gained through acquiring immunological immunity.

The multiple regression analysis in the presented study showed a significant relationship between the number of COVID-19 vaccine doses administered and the impact of the pandemic on work productivity. It was shown that receiving the second and third doses of the vaccine played a significant role in reducing the pandemic’s impact on employee productivity compared to unvaccinated individuals. This indicates that receiving the vaccine reduces and alleviates the course of the coronavirus disease in workers, leading to no reduction in work productivity, thus limiting the costs of lost productivity. Similarly, a comparison of respondents vaccinated with two doses indicated a lower impact of the COVID-19 pandemic on work productivity compared to those vaccinated with only one dose. The Centers for Disease Control and Prevention (CDC) [24] proved that the second dose of the COVID-19 vaccine increases protection efficacy and extends its duration. However, no effect of the type of vaccine received on work productivity during the pandemic was observed.

The number of days of work absence due to COVID-19 infection was also determined by the respondents. Its dependence on the type and number of COVID-19 vaccine doses received was analyzed. This allowed for the observation of a significant relationship indicating that receiving the third vaccine dose substantially reduced the number of work absences compared to unvaccinated individuals and those who received fewer doses. Respondents who completed the full vaccination cycle and received the booster dose had significantly fewer work absence days due to COVID-19, which proves the effectiveness of the vaccine. The National Institute of Public Health (PZH) [25] defines the goals of administering the booster dose as increasing the level of immunological immunity and extending its duration.

The effectiveness of administering the third dose as a booster against COVID-19 was confirmed in the study conducted by Munro et al. [26]. Similar results were presented by Moreira [27], who hypothesized that the primary vaccination series (1 + 2) protects patients from severe disease, hospitalization, and death at a high level, but the third dose plays a crucial role in reducing workplace absences. However, no impact of the type of vaccine received on work absenteeism during the pandemic was observed. Preventive actions enable a reduction in morbidity and absenteeism among workers, allowing employers, and society at large, to avoid the costs associated with the coronavirus disease.

The results align with the study conducted by Maltezou et al. [28] on healthcare workers in Greece, which documented that the average duration of work absences for unvaccinated individuals was higher than for vaccinated individuals (using the BioNTech vaccine).

In Poland, the COVID-19 vaccination program was introduced at the end of 2020, and progressively, with time, different age and occupation-based groups were vaccinated. According to a report presented by the Social Insurance Institution on sick leave in 2020, in the year when Polish society faced COVID-19 without access to vaccines, there were 5068.6 thousand sick leave days issued due to COVID-19. In a similar report prepared in 2021, ZUS recorded 4244.8 thousand sick leave days caused by COVID-19, indicating a decrease in the number of sick leave days by 16.3% compared to the previous year. Based on these data, one can infer that the general access to COVID-19 vaccines and their effectiveness in eliminating and mitigating the disease contributed to the reduction in costs related to employee absenteeism [7].

Assessing the impact of COVID-19 vaccinations (Pfizer/BioNTech, Moderna, Oxford/AstraZeneca, Johnson & Johnson) on selected economic aspects, the above data highlight the rationality of implementing the SARS-CoV-2 vaccination program in Poland. Further research is necessary to enable a proper assessment of the program from a long-term perspective. The presented results are of an innovative nature, providing the first scientific data in Poland regarding the economic evaluation of the COVID-19 vaccination program. 

Moreover, our findings can be used in a global analysis of the effectiveness of the most commonly administered vaccines against COVID-19. Therefore, the presented study meets the information needs of decision-makers and healthcare professionals. Our results may constitute the basis for the developing clear recommendations regarding vaccination against SARS-CoV-2. 

What is more, our results may help to solve the problem of vaccine acceptance among healthcare workers. We clearly showed the benefits of being vaccinated against SARS-CoV-2. If there is no willingness to get vaccinated and no promotion of COVID-19 vaccines, the SARS-CoV-2 pandemic will never end.

However, the presented study has several key limitations. The first one is undoubtedly the online nature of the research, where the main limitation is the willingness of the survey participants to take part in the study. This limitation determines the size of the study group. Additionally, in the study model, we restricted ourselves solely to surveying healthcare workers. This limitation arises from the fact that healthcare workers were the first to be vaccinated against COVID-19 according to the vaccination program, playing a crucial role in the fight against the pandemic. At the same time, in our belief, the surveyed group had the appropriate knowledge and qualifications for a thorough assessment of the effectiveness of the COVID-19 vaccination program, not only from a medical but also from an economic perspective.

## 5. Conclusions

Our results clearly show that the widely implemented national COVID-19 vaccination program brings tangible benefits both in medical and economic terms. The extent of reducing absenteeism and presenteeism caused by the coronavirus disease depended on the number of vaccine doses administered. Receiving a booster dose, regardless of its type, significantly reduced absenteeism caused by COVID-19 among the participants. Further studies are necessary to enable a proper assessment of the program from a long-term perspective.

## Figures and Tables

**Table 1 vaccines-12-00023-t001:** Characteristics of the surveyed respondents (n = 736).

Variables	Respondents		*p* Value
	Unvaccinated respondents	Vaccinated respondents	
Group size n (%)	215 (29.21%)	521 (70.79%)	
Gender (%): Women Men	53.02 46.98	52.97 47.03	0.6623 *
Age Median [Q_1_–Q_3_] ^&^	43 [25–55]	44 [27–56]	0.3578 ^#^
Place of residence (%): village city up to 25,000 inhabitants city from 25 to 100 thousand inhabitants city with more than 100,000 inhabitants city with more than 500,000 inhabitants	27.91 16.74 13.95 18.60 22.80	25.14 11.90 13.05 16.89 33.02	<0.0001 *
Education (%): Secondary Higher	40.00 60.00	38.00 62.00	0.1521 *

Legend: ^&^ median (Q1 [lower quartile]–Q3 [upper quartile]); *—The chi-square for independence; ^#^—The U Mann–Whitney test.

**Table 2 vaccines-12-00023-t002:** The impact of the COVID-19 pandemic on work performance concerning the number of vaccine doses received among the respondents.

The Number of Doses of the COVID-19 Vaccine Received	The Impact of the COVID-19 Vaccines on Work Performance		
	Median	[Q_1_–Q_3_]	*p* Value
0 (Unvaccinated respondents)	3 ^b^	[2–5]	*p* < 0.01 ^#^
1	3 ^b^	[1–5]	
2	2.0 ^a^	[1–4]	
3	1.5 ^a^	[1–3]	

Legend: ^#^—The Kruskal–Wallis test; ^a,b^—groups followed by the same letters do not differ statistically significantly.

**Table 3 vaccines-12-00023-t003:** The impact of the COVID-19 pandemic on work performance concerning the type of received COVID-19 vaccine.

Type of Received COVID-19 Vaccine	The Impact of the COVID-19 Vaccines on Work Performance		
	Median	[Q_1_–Q_3_]	*p* Value
BNT162 b2 Pfizer/BioNTech,	2.5	[1–3]	
Ad26.COV2-S Johnson & Johnson	2.0	[1–4]	
ChAdOx1 nCoV-19 Oxford/AstraZeneca	2.0	[1–3]	
mRNA-1273 Moderna	2.0	[1–3]	0.1313 ^#^

Legend: ^#—^The Kruskal–Wallis test.

**Table 4 vaccines-12-00023-t004:** The number of workdays missed (over the past twelve months) due to COVID-19-related sick leave in relation to the number of doses of the COVID-19 vaccine received.

Number of Doses of the COVID-19 Vaccine Received	The Number of Workdays Missed Due to Sick Leave Related to a Coronavirus Infection		
	Median	[Q_1–_Q_3_]	*p* Value
0 ((Unvaccinated respondents)	5.32 ^b^ 6	±16.24 [5–23]	
1	5.24 ^b^ 5	±21.82 [2–28]	
2	4.12 ^b^ 4	±11.05 [2–15]	
3	2.00 ^a^ 2.5	±6.75 [1–10]	0.0265 ^#^

Legend: ^#^—The Kruskal–Wallis test; ^a,b^—groups followed by the same letters do not differ statistically significantly.

**Table 5 vaccines-12-00023-t005:** The number of workdays missed (over the past twelve months) due to sick leave related to a COVID-19 infection in relation to the type of COVID-19 vaccine received.

Type of COVID-19 Vaccine Received	The Number of Workdays Missed Due to Sick Leave Related to a COVID-19 Infection		
	Median	[Q_1_–Q_3_]	*p* Value
BNT162 b2 Pfizer/BioNTech,	5	[2–16]	
Ad26.COV2-S Johnson & Johnson	2.5	[1–8]	
ChAdOx1 nCoV-19 Oxford/AstraZeneca	3	[1–8]	0.6531 ^#^
mRNA-1273 Moderna	2	[1–8]	

Legend: ^#^—The Kruskal–Wallis test.

## Data Availability

The datasets used and/or analyzed during the current study are available from the corresponding author on reasonable request.

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
