# Peer review of "Impact of the SARS-CoV-2 Vaccination Program on Presenteeism and Absenteeism among Healthcare Workers in Poland"

_vaccines, 2023, doi:10.3390/vaccines12010023_

Round 1

Reviewer 1 Report

Comments and Suggestions for Authors

Exclusion criterion Line 135 , not being vaccinated against COVID-19  [please check and remove]. As you have a control group who are not vaccinated.

You use Kruskal-Wallis H test, this is fine for medians but you are showing means in the tables. Can you check the results with Student t-tests for means, (multivariate tests)?

Author Response

Exclusion criterion Line 135 , not being vaccinated against COVID-19  [please check and remove]. As you have a control group who are not vaccinated.

You use Kruskal-Wallis H test, this is fine for medians but you are showing means in the tables. Can you check the results with Student t-tests for means, (multivariate tests)?

Response:  We corrected presented results into medians and interquartile ranges. The  choice of Kruskal-Wallis  test was correct since data did not followed the normal distribution. I believe it would incorrect to perform Student t-tests pairwise and then make  the corrections for multiple comparisons as we  could just perform ANOVA. However the lack of normality leads to differences between the medians and means (skewed distributions) which would interfere the results. 

Reviewer 2 Report

Comments and Suggestions for Authors

Karolina Hoffmann et. al. present in their manuscript a study how the Covid-19 vaccination influenced the number of working days affected by a Covid-19 infection amongst Polish health care workers. 

The outcome is that the more Covid-19 vaccination (between 0 and 3) an employee received the smaller the number days he or she was absent from work or considerably affected in its working efficiency by a Covid-19 infection (see tables 2 and 4). The type of vaccination had no statistically significant effect (see tables 3 and 5). It is of note that the standard deviation for days missed due to sick leave became significant smaller when the recipient had more than one vaccination - in agreement with the observation that a repeated vaccination not primarily reduces the risk of infection but the severity of the disease. 

I think the data presented in the manuscript support the authors conclusion. Even though the effect of a vaccination on the number of days absent from work is not overwhelming a measurable effect is detectable. 

I think the article can be published as it is. 

Author Response

Karolina Hoffmann et. al. present in their manuscript a study how the Covid-19 vaccination influenced the number of working days affected by a Covid-19 infection amongst Polish health care workers. 

The outcome is that the more Covid-19 vaccination (between 0 and 3) an employee received the smaller the number days he or she was absent from work or considerably affected in its working efficiency by a Covid-19 infection (see tables 2 and 4). The type of vaccination had no statistically significant effect (see tables 3 and 5). It is of note that the standard deviation for days missed due to sick leave became significant smaller when the recipient had more than one vaccination - in agreement with the observation that a repeated vaccination not primarily reduces the risk of infection but the severity of the disease. 

I think the data presented in the manuscript support the authors conclusion. Even though the effect of a vaccination on the number of days absent from work is not overwhelming a measurable effect is detectable. 

I think the article can be published as it is. 

Our comment:

Thank you for reviewing our manuscript.

Reviewer 3 Report

Comments and Suggestions for Authors

Here are some suggestions and potential areas of improvement:

  1. Sampling and Representation: Ensure that the sample size is sufficiently large and representative of the broader population of healthcare workers. Consider including participants from a wider range of healthcare settings to enhance generalizability.

  2. Control for Confounding Variables: The study should control for potential confounding factors that might influence presenteeism and absenteeism, such as workplace policies, personal health status, or external stress factors.

  3. Longitudinal Design Consideration: If possible, employ a longitudinal study design to better understand the long-term effects of vaccination on work performance.

  4. Diversity of Vaccines: Analyze if different types of COVID-19 vaccines have varying impacts on work performance. This could offer deeper insights into the effectiveness of specific vaccines.

  5. Psychological Impact Assessment: Include measures to assess the psychological impact of vaccination on healthcare workers, as this can influence their perception of work performance and overall well-being.

  6. Economic Impact Analysis: Expand the analysis to include a more comprehensive assessment of the economic impact of vaccination, taking into account factors like healthcare costs and productivity losses.

  7. Broader Societal Implications: Discuss the broader implications of your findings on public health policies and vaccination strategies.

  8. Limitations and Future Research: Clearly outline the limitations of your study and suggest areas for future research to build upon your findings.

  9. Ethical Considerations: Ensure that all ethical guidelines and considerations, especially regarding participant consent and data confidentiality, are rigorously followed and clearly documented.

Comments on the Quality of English Language

Here are some suggestions and potential areas of improvement:

  1. Sampling and Representation: Ensure that the sample size is sufficiently large and representative of the broader population of healthcare workers. Consider including participants from a wider range of healthcare settings to enhance generalizability.

  2. Control for Confounding Variables: The study should control for potential confounding factors that might influence presenteeism and absenteeism, such as workplace policies, personal health status, or external stress factors.

  3. Longitudinal Design Consideration: If possible, employ a longitudinal study design to better understand the long-term effects of vaccination on work performance.

  4. Diversity of Vaccines: Analyze if different types of COVID-19 vaccines have varying impacts on work performance. This could offer deeper insights into the effectiveness of specific vaccines.

  5. Psychological Impact Assessment: Include measures to assess the psychological impact of vaccination on healthcare workers, as this can influence their perception of work performance and overall well-being.

  6. Economic Impact Analysis: Expand the analysis to include a more comprehensive assessment of the economic impact of vaccination, taking into account factors like healthcare costs and productivity losses.

  7. Broader Societal Implications: Discuss the broader implications of your findings on public health policies and vaccination strategies.

  8. Limitations and Future Research: Clearly outline the limitations of your study and suggest areas for future research to build upon your findings.

  9. Ethical Considerations: Ensure that all ethical guidelines and considerations, especially regarding participant consent and data confidentiality, are rigorously followed and clearly documented.

Author Response

Here are some suggestions and potential areas of improvement:

  1. Sampling and Representation: Ensure that the sample size is sufficiently large and representative of the broader population of healthcare workers. Consider including participants from a wider range of healthcare settings to enhance generalizability.

Response: According to the Central Statistical Office (GUS), we have N=333,876 registered employees with the right to practice one of the medical professions.  We  originally assessed the  sample size 1063 which satisfied the margin of error at 3% . But since the responders rate  was poor (736) the sample size required  the assumption that the MOE  is at 4% level ( for this level the required sample size is 599). Assumptions for sample size  determination are presented in the part ‘statistical analysis’.

  1. Control for Confounding Variables: The study should control for potential confounding factors that might influence presenteeism and absenteeism, such as workplace policies, personal health status, or external stress factors.

Response: The primary outcome of our study was to assess absenteeism and presenteeism in the context of COVID-19 vaccine uptake. We asked our respondents how many days they were absent from work due to COVID-19 infection. Due to the state of the epidemic, workplace policies have been unified in Polish healthcare facilities. In order to control the spread of the virus, it was necessary to implement preventive sanitary measures in each country. In Poland, the state of the epidemic was declared by the Minister of Health on March 20, 2020. Specific restrictions based on the relevant regulations of the Council of Ministers were introduced. In our future projects, we will take into account the confounding factors mentioned above (workplace policies, personal health status, external stress factors). However, we explored many problems mentioned above in our previous studies – see:

Hoffmann, K., Kopciuch, D., Bońka, A., Michalak, M., Bryl, W., Kus, K., Nowakowska, E., Zaprutko, T., Ratajczak, P., & Paczkowska, A. (2023). The Mental Health of Poles during the COVID-19 Pandemic. International journal of environmental research and public health, 20(3), 2000. https://doi.org/10.3390/ijerph20032000

Hoffmann, K., Paczkowska, A., Bońka, A., Michalak, M., Bryl, W., Kopciuch, D., Zaprutko, T., Ratajczak, P., Nowakowska, E., & Kus, K. (2022). Assessment of the Impact of the COVID-19 Pandemic on the Pro-Health Behavior of Poles. International journal of environmental research and public health, 19(3), 1299. https://doi.org/10.3390/ijerph19031299

Hoffmann, K., Michalak, M., Bońka, A., Bryl, W., Myśliński, W., Kostrzewska, M., Kopciuch, D., Zaprutko, T., Ratajczak, P., Nowakowska, E., Kus, K., & Paczkowska, A. (2023). Association between Compliance with COVID-19 Restrictions and the Risk of SARS-CoV-2 Infection in Poland. Healthcare (Basel, Switzerland), 11(6), 914. https://doi.org/10.3390/healthcare11060914

Paczkowska, A., Hoffmann, K., Michalak, M., Hans-Wytrychowska, A., Bryl, W., Kopciuch, D., Zaprutko, T., Ratajczak, P., Nowakowska, E., & Kus, K. (2022). Safety Profile of COVID-19 Vaccines among Healthcare Workers in Poland. Vaccines, 10(3), 434. https://doi.org/10.3390/vaccines10030434

  1. Longitudinal Design Consideration: If possible, employ a longitudinal study design to better understand the long-term effects of vaccination on work performance.

Response: Thank you for this valuable suggestion. In the presented study, we focused on the period when the willingness to receive the Covid-19 vaccine was the highest. It is really difficult to study the long-term impact of Covid-19 vaccinations on work performance because these vaccinations need to be repeated. After administration of a single dose of the vaccine, antibodies are detected in human serum for up to 6-8 months [Haq]. We addressed our questionnaire to respondents who took COVID-19 vaccine during last 3 months.

Haq, M., & Deshpande, S. V. (2023). Effects of Antibodies in the Serum After the Administration of COVID Vaccines and Their Hematological and Cardiovascular Complications. Cureus, 15(10), e47984. https://doi.org/10.7759/cureus.47984

  1. Diversity of Vaccines: Analyze if different types of COVID-19 vaccines have varying impacts on work performance. This could offer deeper insights into the effectiveness of specific vaccines.

Response: As we showed in the Results section, the type of COVID-19 vaccine received did not have a significant impact (p=0.6531) on the number of workdays missed (over the past twelve months) due to sick leave related to COVID-19 infection among the respondents. The number of workdays missed due to COVID-19 sick leave was lowest in the group vaccinated with the Moderna vaccine and highest in the group vaccinated with the Pfizer/BioNtech vaccine. In the study period there were four vaccines available in Poland: two mRNA vaccines (Pfizer/BioNTech & Moderna) and two vector vaccines (Oxford/AstraZeneca, Johnson&Johnson). However, all of them had their specific effectiveness in preventing SARS-CoV-2 infection, which could translate into various effects on work performance.

  1. Psychological Impact Assessment: Include measures to assess the psychological impact of vaccination on healthcare workers, as this can influence their perception of work performance and overall well-being.

Response: In the presented scientific project, we did not focus on the assessment of mental health, including the psychological impact of vaccination on our respondents. However, in our previous research, we have examined many aspects of mental well-being during the pandemic – see:

Hoffmann, K., Kopciuch, D., Bońka, A., Michalak, M., Bryl, W., Kus, K., Nowakowska, E., Zaprutko, T., Ratajczak, P., & Paczkowska, A. (2023). The Mental Health of Poles during the COVID-19 Pandemic. International journal of environmental research and public health, 20(3), 2000. https://doi.org/10.3390/ijerph20032000

Hoffmann, K., Paczkowska, A., Bońka, A., Michalak, M., Bryl, W., Kopciuch, D., Zaprutko, T., Ratajczak, P., Nowakowska, E., & Kus, K. (2022). Assessment of the Impact of the COVID-19 Pandemic on the Pro-Health Behavior of Poles. International journal of environmental research and public health, 19(3), 1299. https://doi.org/10.3390/ijerph19031299

Hoffmann, K., Michalak, M., Bońka, A., Bryl, W., Myśliński, W., Kostrzewska, M., Kopciuch, D., Zaprutko, T., Ratajczak, P., Nowakowska, E., Kus, K., & Paczkowska, A. (2023). Association between Compliance with COVID-19 Restrictions and the Risk of SARS-CoV-2 Infection in Poland. Healthcare (Basel, Switzerland), 11(6), 914. https://doi.org/10.3390/healthcare11060914

Paczkowska, A., Hoffmann, K., Michalak, M., Hans-Wytrychowska, A., Bryl, W., Kopciuch, D., Zaprutko, T., Ratajczak, P., Nowakowska, E., & Kus, K. (2022). Safety Profile of COVID-19 Vaccines among Healthcare Workers in Poland. Vaccines, 10(3), 434. https://doi.org/10.3390/vaccines10030434

  1. Economic Impact Analysis: Expand the analysis to include a more comprehensive assessment of the economic impact of vaccination, taking into account factors like healthcare costs and productivity losses.

Response: We appreciate the valuable suggestion from the reviewer to enhance the scientific merit of our paper. We will certainly take the above advice into consideration when designing future scientific research projects. It is worth emphasizing that, in the opinion of many health economics experts, the primary limitation in calculating the direct and especially indirect costs of specific diseases is the access to data. In Poland, there is still a lack of a sufficient number of well-maintained public registries of patients. The National Health Fund is the payer for medical procedures, while the Social Insurance Institution is the payer for social benefits in Poland. Neither of these institutions discloses the costs generated by an individual patient; they only provide general data for a specific disease entity.

According to the latest public data on expenditures from the Covid-19 Prevention Fund estimated by the Supreme Audit Office for the years 2020-2022, these expenditures in the healthcare sector amounted to over 13 million Euros, and in the social insurance sector, they exceeded 7 million Euros (https://www.nik.gov.pl/aktualnosci/fundusz-przeciwdzialania-covid-19.html).  According to a report presented by the Social Insurance Institution on sick leave in 2020, the year when Polish society faced COVID-19 without access to vaccines, there were 5,068.6 thousand sick leave days issued due to COVID-19. In a similar report prepared in 2021, ZUS recorded 4,244.8 thousand sick leave days caused by COVID-19, indicating a decrease in the number of sick leave days by 16.3% compared to the previous year. Based on this data, one can infer that the general access to COVID-19 vaccines and their effectiveness in eliminating and mitigating the disease contributed to the reduction of costs related to employee absenteeism. Therefore, the results of our research are innovative, representing one of the few scientific reports confirming the positive impact of the Covid-19 vaccination program on social aspects (presenteeism and absenteeism). Consequently, they confirm the cost-effectiveness of this medical technology in the context of public health. Above information is included in the discussion section of our paper.

  1. Broader Societal Implications: Discuss the broader implications of your findings on public health policies and vaccination strategies.

Response: The results of our research are innovative, representing one of the few scientific reports confirming the positive impact of the Covid-19 vaccination program on social aspects (presenteeism and absenteeism). Consequently, they confirm the cost-effectiveness of this medical technology in the context of public health. In the future, our findings may be one of the arguments for vaccination against COVID-19, as one of the factors responsible for the success of vaccination campaigns is public willingness to be vaccinated. Our data highlighted the role of healthcare workers as role models and an important source of vaccination knowledge for the rest of society. The need for continued research and vigilance in monitoring vaccine effectiveness, especially as variants emerge and as vaccination strategies evolve, must be emphasized. More and more data are still needed to demonstrate the efficacy of COVID-19 vaccines in specific groups of recipients, including healthcare professionals.

Findings from this research are innovative and can be used in a global analysis of the effectiveness of the most commonly administered vaccines against COVID-19. Therefore, the presented study meets the information needs of decision-makers and healthcare professionals. Our results may constitute the basis for the developing clear recommendations regarding vaccination against SARS-CoV-2.

What is more, our results may help to solve the problem of vaccine acceptance among healthcare workers. We clearly showed the benefits of being vaccinated against SARS-CoV-2. If there is no willingness to get vaccinated and no promotion of COVID-19 vaccines, the SARS-CoV-2 pandemic will never end. Above information is included in the discussion section of the paper.

  1. Limitations and Future Research: Clearly outline the limitations of your study and suggest areas for future research to build upon your findings.

Response: Presented study has several key limitations. The first one is undoubtedly the online nature of the research, where the main limitation is the willingness of the survey participants to take part in the study. This limitation determines the size of the study group. Additionally, in the study model, we restricted ourselves solely to surveying healthcare workers. This limitation arises from the fact that healthcare workers were the first to be vaccinated against Covid-19 according to the vaccination program, playing a crucial role in the fight against the pandemic. At the same time, in our belief, the surveyed group had the appropriate knowledge and qualifications for a thorough assessment of the effectiveness of the Covid-19 vaccination program, not only from a medical but also from an economic perspective. Above information is included in the discussion section of our paper.

  1. Ethical Considerations: Ensure that all ethical guidelines and considerations, especially regarding participant consent and data confidentiality, are rigorously followed and clearly documented.

Response: The survey process began following an opinion  (No. 25/22) issued by the Bioethics Committee at the Poznan University of Medical Sciences confirming that the study does not have the features of a medical experiment. Before taking the survey, each participant was made aware of the purpose of the research and was informed that the study was safe, free, and anonymous, and that the consent to participate in the research could be withdrawn at any time, without giving any reason. All obtained information was treated as fully confidential, and the identity of all participants remained anonymous. No personal data of respondents were collected, ensuring compliance with the Personal Data Protection Act, as no data allowing subsequent identification of participants were used in the research project. Above information is included in the ‘material and methods’ section of the paper.

Round 2

Reviewer 3 Report

Comments and Suggestions for Authors

The revisions made to the article were found sufficient.

Comments on the Quality of English Language

The revisions made to the article were found sufficient.